# The Derived Components of *Gnaphalium hypoleucum* DC. Reduce Quorum Sensing of *Chromobacterium violaceum*

**DOI:** 10.3390/molecules27154881

**Published:** 2022-07-30

**Authors:** Yu-Long Li, Zi-Yong Chu, Gui-Min Liu, Sheng-Qiang Yang, Hong Zeng

**Affiliations:** 1School of Basic Medicine, Youjiang Medical University for Nationalities, Baise 533000, China; yulongli128@163.com (Y.-L.L.); 19177603925@163.com (S.-Q.Y.); 2Key Laboratory of Protection and Utilization of Biological Resources in Tarim Basin of Xinjiang Production & Construction Corps, College of Life Science and Technology, Tarim University, Alar 843301, China; chuziyong0312@gmail.com (Z.-Y.C.); lgm1786368111@163.com (G.-M.L.)

**Keywords:** quorum sensing, *Gnaphalium hypoleucum* DC., molecular docking, *Chromobacterium violaceum* ATCC 12472, biofilm

## Abstract

*Gnaphalium hypoleucum* DC. was first recorded in the Chinese National Pharmacopoeia “Yi Plant Medicine”. There is no detailed report on its main components’ activity in suppressing the quorum sensing activity (QS) of bacteria. Our study aimed to screen the main components in extracts of *G. hypoleucum* DC. in order to measure their effects on bacterial QS activity and to explore specific quorum sensing mechanisms that are affected by *G**. hypoleucum* DC. extracts. Crude extracts of *G. hypoleucum* DC. contained significant amounts of two compounds shown to inhibit bacterial QS activity, namely apigenin and luteolin. Apigenin and luteolin in crude extracts of *G. hypoleucum* DC. showed substantial inhibition of pigment formation, biofilm production, and motility in *Chromobacterium violaceum* ATCC 12472 compared to the effects of other phytochemicals from *G. hypoleucum* DC. Apigenin and luteolin exhibited a strong QS inhibitory effect on *C. violaceum*, interfering with the violacein pigment biosynthesis by downregulating the *vioB*, *vioC*, and *vio**D* genes. In the presence of signal molecules, the QS effect is prevented, and the selected compounds can still inhibit the production of the characteristic purple pigment in *C. violaceum*. Based on qualitative and quantitative research using genomics and bioinformatics, we concluded that apigenin and luteolin in crude extracts of *G. hypoleucum* DC can interfere with the generation of QS in *C. violaceum* by downregulating the *vioB*, *vioC*, and *vio**D* genes. Indeed, *G. hypoleucum* DC. is used for the treatment of bacterial infections, and this research provides new ideas and potential alternative uses for medicinal plants.

## 1. Introduction

Quorum sensing (QS) is a cell-to-cell communication pathway or process which is universal in fungi and bacteria, especially in many human opportunistic pathogens. It regulates many physiological and biochemical functions, including antibiotics, bacteriocin production, pathogenesis, biofilm formation, conjugation, competence, etc. [1]. The literature reported that interfering with QS may be a promising alternative strategy for controlling bacterial infections, especially in antibiotic-resistant strains [2]. *Chromobacterium violaceum* typically produces the pigment violacein, controlled by QS. In addition, *C. violaceum*, as a gram-negative opportunist, is a commonly used bacterial species for QS research, and it is also a candidate for detecting a decrease in the secretion of virulence factors when the QS system is disrupted. Quorum sensing inhibitors (QSIs), which prevent violacein production by this pathogen without any bacterial inhibition, are capable of attenuating bacterial pathogenicity with a lower risk of promoting resistance compared to that associated with the use of antibiotics. Previous papers have reported that the essential oil phytol, sponge extracts, okanin, and amicoumacins exhibit anti-quorum sensing activity [3,4,5,6]. Interestingly, plant extracts have also been reported to manifest potential QS inhibition, which may offer opportunities for weakening the virulence of pathogens, especially in multidrug-resistant bacteria. Thus, active plants extracts are alternative screening targets for QSIs. The *G**. hypoleucum* DC., belongs to the genus *Gnaphalium* in the family Compositae, which widely distributed in the Yunnan province of China. The plant is usually used as a wild vegetable, named “Qing Ming Cai” in China, which can be used to make the ‘‘Qing Ming Guo” dish. It also has antipyretic, anti-inflammatory, antitussive, anti-gout, and expectorant activities [7]. Additionally, it is reported that two flavonoids from *G. hypoleucum* DC. showed xanthine oxidase (XO) inhibitory activity [8]. Phytochemical compounds in the plant are, for the most part, flavonoids, and most of them are 5-OH flavanones, aurantiamide acetate, 5-hydroxy-3,6,7,8-tetramethoxyflavone, 5-hydroxy-3,6,7,8,4′-pentamethoxy flavones, tetracosanoic acid, β-sitosterol,5-hydroxy-3,6,7,8,3′,4′-hexa methoxyflavone, 5,7-dihydroxy-3,8,4′-trimethoxyflavone, and 5,8-Dihydroxy-3,6,7-trimethoxyflavone [9,10].

Considering the many biological functions reported for extracts of *G. hypoleucum* DC., we designed further research with the aim of evaluating the effects of *G. hypoleucum* DC. extracts on QS activity and bacterial biofilm formation. We obtained and analyzed specific chemical constituents in extracts of *G. hypoleucum* DC. using column chromatography and metabonomics, respectively. Based on the activity tracking system, different crude fractions of *G. hypoleucum* DC. extracts and active compounds were screened for anti-QS activity. Using qRT–PCR and molecular docking, active compounds were then assessed for anti-QS and anti-biofilm activity, as well as for swimming and swarming interference, to obtain further insight regarding the specific mechanisms of action.

## 2. Results 

### 2.1. Metabolome Identification Results of Gnaphalium hypoleucum DC.

Metabolome is a relatively accurate method for identifying secondary metabolites in plants. In this paper, metabolome is used to accurately identify chemical components in *G. hypoleucum* DC. Whole plants were divided into above-ground and underground parts, and the stems, leaves, and flowers were selected as the above-ground parts (DS), while the roots were selected as the underground parts (DX). Three biological replicates were set in each group, with one group being made up of the above-ground parts, and the other being made up of the underground parts. The HCA (hierarchical cluster analysis) results showed that there were significant differences between the DS and DX samples, and the difference between the two groups was small (Figure 1a). The PCA results showed that there were three separate groups of flavonoid metabolites, and the groups were aggregated. The two main components, namely PC1 and PC2 were found to have values of 86.64% and 7.02%, respectively, of which MIX is the quality control sample. Furthermore, the accumulation mode of flavonoids in different parts of *G**. hypoleucum* DC. is also different (Figure 1b). Pearson correlation coefficients (PCC) between samples were calculated using the cor function in R, and were presented as samples, while metabolites were presented as heatmaps with dendrograms (Figure 1c). For HCA, normalized signal intensities of metabolites (unit variance scaling) are visualized as a color spectrum. Metabolome results indicated that apigenin, luteolin, 5,7-dihydroxy-3-methoxyflavone, aromadendrin-7-O-glucoside, peonidin-3-O-glucoside, kaempferol, okanin, quercetin, and flavonoid components are identified in the DS and DX parts of *G. hypoleucum* DC. It is worth noting that the apigenin and luteolin content are higher in the DS part of *G. hypoleucum* DC.

### 2.2. Minimum Inhibitory Concentration of Crude Extracts and Active Compounds against C. violaceum

It is interesting that compounds **1**–**5** were flavonoid monomers and, when finally identified, their structures were luteolin, kaempferol, quercetin, rutin, and apigenin (Appendix A). Minimum inhibitory concentration (MIC) was determined for each of crude extracts and compounds at concentrations ranging from 2000 to 3.91 µg/mL against *C. violaceum*. The MIC value of crude extracts, fraction EE, and rutin were found to be ≥1000 µg/mL, while apigenin, luteolin, kaempferol, and quercetin were found to be 500 µg/mL (Appendix A).

### 2.3. Active Compounds from G. hypoleucum DC. Shown an Inhibitory Effect on Violacein Production of C. violaceum

The synthesis of violacein is regulated by the QS system, which usually used as a simple and intuitive indicator for screening QSIs [11]. Crude extracts from *G. hypoleucum* DC were tested in terms of their violacein inhibition of *C. violaceum*, and the phenomenon shown in (Figure 2) suggested that there was some inhibitory substance(s) in crude extracts, the EE fraction, and compounds from *G. hypoleucum* DC. Compounds **1**–**5**, which were isolated from the EE fraction through repeated column chromatography, exhibited a higher inhibitory effect on violacein production than the EE fraction, which means that they were the main anti-QS activity compounds in EE fraction. The tested crude extracts, EE fraction, apigenin, and luteolin could reduce purple pigment production in *C. violaceum* with an inhibition rate of 30.01%, 39.62%, 58.31%, and 53.78%, respectively, at a concentration of 125 µg/mL, while no interference with the growth of *C. violaceum* was observed. Results showed that the QS inhibitory effects of flavonoid compounds isolated from *G. hypoleucum* DC. could be ranked in the following order: apigenin > luteolin > kaempferol > quercetin > rutin. From this, apigenin and luteolin with better activity were screened out. 

### 2.4. Anti-Biofilm Activity of Luteolin and Apigenin on C. violaceum 

The formation of bacterial biofilms is beneficial to enhance tolerance to antibiotics, and environmental pressure. The formation of bacterial biofilms is regulated by the QS system [12]. In this study, the biofilm inhibition potential of EE fraction, apigenin, and luteolin were evaluated at different concentrations. The formation of biofilms increased with decreasing drug concentration in a classic dose-dependent response. At a concentration of 31.25 µg/mL, the biofilm inhibition rates were 46.15%, 59.05%, and 78.97% for apigenin, luteolin, and EE fraction, respectively (Figure 3). These results demonstrate a significant ability to inhibit the formation of biofilms. In summary, the degree of inhibition of biofilms was measured as apigenin > luteolin, and both compounds were identified as having substantial anti-biofilm and anti-QS activity.

To further verify that apigenin and luteolin are effective chemical components in the EE fraction for inhibiting biofilm formation, we chose to re-evaluate the inhibitory effects on biofilm formation using SEM, because the formation of biofilms is a multistage process. First, bacteria adhere to a surface, such as a glass slide. Then, extracellular proteins, extracellular DNA, and extracellular polysaccharides are produced and accumulated. Different concentrations of the EE fraction, apigenin, and luteolin were added to a culture medium system to observe their effects on biofilm formation (Figure 4). The results showed that *C. violaceum* grows at a very high density and forms a tightly organized biofilm with cells closely adhered to each other in the control. When increasing the concentrations of apigenin or luteolin, the distance between cells in the biofilm increases, the number of dense colonies decreases, and the overall population density is significantly reduced. The spacing between biofilm cells in the apigenin group was greater than that in the luteolin group. The verification results of the *C. violaceum* pigment production inhibition experiment and biofilm formation inhibition experiment were identical.

### 2.5. Reduced Swarming Activity of Luteolin and Apigenin on C. violaceum

Bacterial swarming is beneficial to cells, as they migrate from the inoculation point to the surrounding area; clustering plays a key role in the initial stages of QS-regulated bacterial biofilm formation [13]. Our results show that apigenin and luteolin inhibited the swarm movement behavior of *C. violaceum* in a concentration-dependent manner. Quantitative data showed that swimming distances were 75 mm in the control group, but that swimming distances were 18.06 ± 2.21, 23.94 ± 1.81, and 28.21 ± 2.74 after treatment with apigenin at 15.63, 7.81 µg/mL, and 3.91 µg/mL, respectively. Meanwhile, swimming distances were 23.12 ± 2.42, 25.73 ± 2.21, and 30,06±3.14 after treatment with luteolin at 31.25, 15.63, and 7.81 μg/mL respectively. Qualitative data showed that the swimming distances of *C. violaceum* were obviously reduced after treatment with apigenin and luteolin at different concentrations compared with the control group. At the same time and concentration, we observed that apigenin and luteolin obviously inhibited swimming (Figure 5).

### 2.6. Molecular Docking Prediction

To further analyze the mechanism of action of luteolin and apigenin, potential binding interactions between these two compounds and *vioA*, *vioB*, *vioC*, *vioD*, and *vioE* were conducted through molecular docking. To judge their binding difficulty, we attempted to calculate cell binding energy by using PyMOL software to integrate the angles displayed between the binding site and binding group of the drug molecule and the target protein. We made a 3D map to show the docking site and configuration (Appendix A). The results indicated that the main component of apigenin had docking scores of −6.08 kcal/mol, −5.31 kcal/mol, −6.34 kcal/mol, and −6.83 kcal/mol for *vioA*, *vioB*, *vioC*, and *vioD*, respectively. Meanwhile, the main component of luteolin had docking scores of −5.81 kcal/mol, −4.59 kcal/mol, −6.28 kcal/mol, and −6.49 kcal/mol for *vioB*, *vioC*, and *vioD*, respectively (Figure 6). Hydrogen interactions between *vioC* and apigenin were found in residues CYS239, ARG225, PHE314, PHE213, and LEU315. Hydrogen interactions between *vioD* and apigenin were found in residues VAL45, MSE 175, TYR192, SER277, and ARG353 (Figure 7). Hydrogen interactions between *vioC* and luteolin were found in residues ALA223, ALA237, TYR403, and ASN365. Hydrogen interactions between *vioD* and luteolin were found in residues LYS33, GLU32, ILE6, and GLY7 (Figure 7). We obtained a similar result showing that the binding energy is apigenin < luteolin, which indicates that the inhibitory gene expression ability is apigenin > luteolin.

### 2.7. Apigenin and Luteolin Inhibits Violacein Production through Influencing the Expression of vioABCDE Operon

Combining the above results, two compounds, apigenin and luteolin, showed the strongest inhibitory influence on violacein production by *C. violaceum* (Figure 8). We investigated further to determine how apigenin and luteolin inhibit violacein production. Studies have reported that five genes, including *vio**A*, *vio**B*, *vio**C*, *vio**D*, and *vioE* are involved in violacein biosynthesis by *C. violaceum* [14]. Thus, the effects of apigenin and luteolin on the expression of *vioA*, *vioB*, *vioC*, and *vioD* in *C. violaceum* were evaluated by qRT–PCR. Our results found that, after treatment of *C. violaceum* with different concentrations of apigenin and luteolin, *vioB*, *vioC*, and *vioD* were significantly downregulated in a dose-dependent manner, but *vioA* was upregulated in a dose-dependent manner. Compared to the untreated control, *vioB*, *vioC*, and *vioD* were downregulated at least 80-fold, 117-fold, and 61.2-fold, respectively, when *C. violaceum* was treated with apigenin at 31.25 μg/mL. Meanwhile, *vioB*, *vioC*, and *vioD* were downregulated at least 47-fold, 69-fold, and 24-fold, respectively, when *C. violaceum* was treated with luteolin at 31.25 μg/mL.

In summary, the inhibitory gene expression ability is apigenin > luteolin, and both can block the quorum sensing of *C. violaceum*.

## 3. Discussion

The MDR pathogen still poses a global threat, and QS arose as a promising strategy to combat MDR strains. The idea is to use QSI agents to disarm pathogens of virulence factors. Thus, active plants extracts are alternative screening targets for QSIs. With this in mind, the chemical composition and inhibition effect of *G. hypoleucum* DC., which has various biological activities, were analyzed. *G. hypoleucum* DC. belongs to *Gnaphalieae*, which is traditionally used as an edible plant, and flavonoid compounds and diterpenes are major active constituents. The total flavonoids from *G. hypoleucum* DC. were analyzed by UHPLC–QTOF/MS [10], while total flavonoids from *G. hypoleucum* DC., including DS and DX, were identified by metabolomics in our research. There are different identification results between UHPLC–QTOF/MS and metabolomics, perhaps because the two methods used different chemical databases. We isolated a total of five flavonoid compounds including apigenin, luteolin, kaempferol, quercetin, and rutin, based on anti-QS activity tracking. Sun Qun et al. (2012) reported that apigenin, luteolin, quercetin, luteolin-4’-O-β-D-glucoside, and quercetin-4’-O-β-D-glucoside were isolated from *G. hypoleucum* DC [15]. Some flavonoid compounds showed antibacterial and anticomplement activities as well as α-glucosidase inhibiting activities in vitro. Our results showed that the MIC values of crude extract fraction EE and rutin were ≥1000 µg/mL, while those of apigenin, luteolin, kaempferol, and quercetin were 500 µg/mL. Bali et al. (2019) reported that the MIC values of luteolin and apigenin isolated from dietary phytochemicals against *C. violaceum* were 750 µg/mL. This result differs from our data, which may be due to the use of different methods employed to detect the MIC values and variations in the preparation of the solutions of phytochemicals [16].

Published reports show that flavonoid compounds, including luteolin, kaempferol, quercetin, and rutin, have demonstrated anti-QS activities [17,18]. In addition, some flavonoid compounds may synergistically reinforce anti-QS activity, such as rutin and apigenin or rutin and naringenin [19]. Our results showed that monomeric compounds, including apigenin, luteolin, kaempferol, quercetin, and rutin, were more potent than ethyl acetate extract (fraction EE). For the first time, anti-QS activity was detected in *G. hypoleucum* DC., and the EE fraction inhibited the production of violacein, a quorum sensing regulated pigment, in a *C. violaceum* tester strain without interfering with its growth.

The formation of biofilms is regulated by QS, and the ability to form biofilm contributes to the pathogenesis of bacterial infection. Anti-biofilm agents can prevent the biofilm mode of growth and result in failure to establish an infection. Furanone and furanone derivative have a good inhibition effect of biofilm formation, such as the food poisoning bacterium *Bacillus cereus*, *Staphylococcus aureus*, *Staphylococcus epidermidis*, and *Micrococcus luteus* [20]. Considering the influence of phytochemicals on the quorum sensing effect of *C. violaceum*, some studies have reported that luteolin and apigenin reduce biofilm formation in *V. harveyi* and *E. coli* O157:H7. In addition, some papers also reported that apigenin was an effective polyphenol in reducing the biofilm formation of *C. albicans* [14,21,22]. Our results further verified that luteolin and apigenin reduced biofilm formation in *C. violaceum*.

Bacterial motility is an important factor in the early stages of biofilm formation and it is also linked to QS. We aimed to assess both types of motility in the presence of either compound derived from *G. hypoleucum* DC. Our results showed a remarkable decrease in the swimming and swarming motility of *C. violaceum* when treated with apigenin and luteolin [23]. However, Bali et al. (2019) reported that apigenin and luteolin from dietary phytochemicals reduced the swimming and swarming behavior of *P. aeruginosa* PAO1 [16]. This may be because apigenin and luteolin showed different sensitivities to *P. aeruginosa* PAO1 and *C. violaceum*, leading to differences in the inhibition of swimming and swarming motility.

Violacein production is an important virulence factor in *C. violaceum*, which is regulated by the CviR protein and five enzyme-encoding genes, namely *vioA*, *vioB*, *vioC*, *vioD*, and *vioE* [24]. Once the expression of one of these five particular genes is changed, the production of purple pigment synthesis is reduced in *C. violaceum*, thus, reducing its toxicity. Our molecular docking results indicated that main component of apigenin and luteolin had a docking score of −5.8 kcal/mol and −5.46 kcal/mol with the CviR protein, respectively (Appendix A). We specifically found that apigenin and luteolin were well-placed in the AHL binding pockets of the CviR protein of *C. violaceum* (Appendix A). Our consideration of apigenin and luteolin for further assessment of virulence activity in *C. violaceum* was based on changes in the activity of the *vioABCDE* genes. In this study, compared with the untreated group, the *vioB*, *vioC*, and *vioD* genes were downregulated, and the *vioA* gene was upregulated in the treatment group, which reduced violacein production of *C. violaceum*, as determined using qRT–PCR and molecular docking predictions. Our previous study of amicoumacins isolated from desert bacteria showed that the expression of *vioA*, *vioD*, and *vioE* was significantly downregulated, and *vioC* was upregulated [25]. The results of our two molecular docking experiments are inconsistent, which may be due to different structural types of compounds and different pathways affecting the synthesis of purple pigment. Furthermore, Xu et al. (2017) reported that the expression of *vio**B*, *vioC*, and *v**ioD* enzymes was more important than other enzymes (*vioA* and *vioE*) during violacein biosynthesis by using a statistical approach for metabolic engineering [26]. Subsequently, Musa Ibrahim et al. (2020) determined that kitasamycin and nitrofurantoin downregulated the *vioB*, *vioC*, and *v**ioD* enzymes [27]. Our qRT–PCR results presented here are consistent with previous reports, which studied the effect of QSIs from plant sources on the expression of QS-related genes [28,29].

## 4. Materials and Methods

### 4.1. Metabolomics Analysis and Active Ingredient Extraction of Gnaphalium hypoleucum DC.

Whole *G. hypoleucum* DC. plants were purchased from a market in the Yunnan province and were identified by Pro. Yanping Liu (NO. YNN20190008). The specimens were then deposited in the School of Basic Medicine, You jiang Medical University for Nationalities. We randomly sampled 1 kg of *G. hypoleucum* DC. with similar phenotypic trait vigor and height. They were divided into two groups of above-ground and underground parts, and the stems, leaves, and flowers were selected as the above-ground parts (DS), while the roots were selected as the underground parts (DX). Three biological replicates were set in each group. Biological samples were freeze-dried using a vacuum freeze-dryer before being crushed using a mixer mill (MM 400, Retsch) with a zirconia bead for 1.5 min at 30 Hz. The lyophilized powder was dissolved in methanol solution, all effective compounds were extracted overnight, and the extract was filtered to remove the remaining lyophilized powder, before the UPLC-MS/MS analysis.

The sample extracts were analyzed using an ultra-performance liquid chromatography electrospray ionization mass spectrometry (UPLC-ESI-MS/MS) system (UPLC, SHIMADZU NexeraX2, www.shimadzu.eom.cn/ (accessed on 16 June 2021); MS, Applied Biosystems 4500 Q TRAP, www.appliedbiosystems.eom.cn/ (accessed on 30 June 2021)). The analytical conditions were as follows: column, Agilent SB-C18 (1.8 μm, 2.1 mm × l00 mm); the mobile phase consisted of solvent A, pure water with 0.1% formic acid, and solvent B, acetonitrile with 0.1% formic acid. The flow velocity was set as 0.35ml per minute; the column oven was set to 40 °C; the injection volume was 4 μL. The effluent was alternatively connected to an ESI-triple quadrupole-linear ion trap (QTRAP)-MS.

The LIT and triple quadrupole (QQQ, Framingham, MA, USA) scans were acquired on a triple quadrupole-linear ion trap mass spectrometer (Q TRAP, Framingham, MA, USA), AB4500 Q TRAP UPLC/MS/MS System, equipped with an ESI Turbo Ion-Spray interface, operating in positive and negative ion mode and controlled by Analyst 1.6.3 software (AB Sciex, Framingham, MA, USA). Instrument tuning and mass calibration were performed with 10 and 100 μmol/L polypropylene glycol solutions in QQQ and LIT modes, respectively. A specific set of MRM transitions were monitored for each period according to the metabolites eluted within this period.

The whole herb were ground to powder and cold soaked in a 90% ethanol solution for 14 days at solid–liquid with 1:20 (*w*/*v*) and repeated four times. Extracting solution concentrated with a rotary evaporator (Heidolph, Schwabach, Germany) was then used.

The crude extract was dissolved in distilled water by ultrasonic vibration. It was then subjected to silica gel column chromatography and eluted with petroleum ether, ethyl acetate, and ethanol to obtain four fractions. Based on active tracking, the positive fraction was further repeated using column chromatography.

Unsupervised PCA (principal component analysis) was performed by the statistics function prcomp within R (www.r-project.org, accessed on 30 June 2022). The data was unit variance scaled before unsupervised PCA. The HCA (hierarchical cluster analysis) results of samples and metabolites were presented as heatmaps with dendrograms, while Pearson correlation coefficients (PCC) between samples were calculated by the cor function in R and presented as heatmaps. Both HCA and PCC were carried out using the R package pheatmap. For HCA, normalized signal intensities of metabolites (unit variance scaling) are visualized as a color spectrum.

### 4.2. Determination of Minimum Inhibitory Concentrations

Minimum inhibitory concentration (MIC) values for the activity of the extracts and compounds from *G. hypoleucum* DC. on *C. violaceum* were determined by the broth microdilution test in U-96-well plates. Briefly, crude extracts and compounds were mixed with 1% dimethyl sulfoxide (DMSO). *C. violaceum* was cultured at 30 °C for 16 h and then diluted 1:100 with fresh LB broth. Then, 100 μL of diluted live cells and 100 μL of different concentrations of crude extracts and compounds (0–2 mg/mL) from the *G. hypoleucum* DC. mixture were cultured at 30 °C for 24 h without shaking. Each experiment included negative (medium, 1% DMSO) and positive controls (medium including the inoculum). The MIC was recorded as the lowest concentration at which there was no visible growth of the bacteria. The MIC assay was repeated at least three times.

### 4.3. Anti-QS Activity of Extracts and Compounds Isolated from Gnaphalium hypoleucum DC.

Crude extracts (0–0.5 mg/mL) and compounds (0–0.25 mg/mL) were tested for the assessment of anti-QS activity. Briefly, crude extracts and compounds from *G. hypoleucum* DC. were dissolved in 1% DMSO. *C. violaceum* was cultured at 30 °C for 16 h and then diluted 1:100 with fresh LB broth. Then, 100 μL of diluted live cells and 100 μL of different concentrations of crude extracts and compounds from the plant were cultured at 30 °C for 16 h without shaking. Quantitative analysis of violacein production was conducted according to the methods reported in the literature [30]. We aspirated 200 µL of the different treatment groups’ culture in a 96-well plate from each culture and then added the same amount of ethyl acetate. The mixture was then shaken at maximum speed by a vortex apparatus for 2 min and centrifuged at 6000 g for 1 min. The organic phase was recovered, and absorbance was read with a spectrophotometer at a wavelength of 575 nm. To calculate the percentage of inhibition, absorbance of the controls was considered to be 100% production of violacein. Each experiment was performed with three independent cultures.

### 4.4. Anti-Biofilm Activity of Crystal Violet Extracts and Compounds Isolated from G. hypoleucum DC.

Anti-biofilm activity was studied in the presence or absence of apigenin and luteolin using a crystal violet assay at 0.01% (*v*/*v*) concentration for *C. violaceum*. Briefly, overnight cultures of *C. violaceum* adjusted with sterile saline solution and a compound–bacterial suspension mixture was added into each well of the 96-well micro-plate and incubated for 24 h at 37 °C. Then, the medium was removed and washed with sterile water three times. The plates were dried at 65 °C in a universal oven and then 200 µL of a 1% m/v aqueous solution of crystal violet (CV) was added. The stain was allowed to fix at room temperature for 20 min, after which the dye was removed from the wells by washing thoroughly with sterile water. Then, 95% ethanol solution was added to the well, and the CV was dissolved at 37 °C for 30 minutes, before being determined at 470 nm [4]. Inhibitor-mediated reduction in biofilm formation was assessed by comparing it to the positive control without phytochemicals. The biofilm assay was performed at least three times.

### 4.5. Swimming and Swarming Assays

We examined the effects of apigenin and luteolin on the clustering movement of *C. violaceum* according to the previously described method. Drugs were added to the colony exercise medium containing 1% tryptone, 0.5% NaCl, 0.5% agar, and 0.5% D-glucose so that the medium contained drug concentrations at 31.25, 15.63, 7.81 μg/mL, and we cultured 5 µL of overnight cultured bacteria in 6 mm filter paper [28]. By measuring the swimming distance of the bacteria, the swimming ability of each group was compared to evaluate the ability of the drug to inhibit the quorum sensing effect.

### 4.6. SEM Observation

In a 6-well cell culture plate, the dietary plant compound and LB were added to yield a series of solutions at 31.25, 15.63, and 7.81 μg/mL, which were inoculated with *C. violaceum*, and incubated at 30 °C for 24 h. These cultures were then aspirated for dehydration using a gradient ethanol and acetone series, and each gradient was processed at an interval of 15 min, with a final critical point drying apparatus. Finally, the samples were electrostatically coated with gold for observation under a scanning electron microscope (SEM—variable vacuum, ultrahigh resolution, field emission scanning electron microscope/energy spectrometer Thermo Fisher Waltham, MA, USA).

### 4.7. Molecular Docking Prediction

Docking analysis was performed using a molecular operating environment (MOE, 2014.09) according to the method of [25]. We developed docking predictions using standard compounds with the CviR (Q7NQP7), vio proteins A (Q9S3V1), B (Q9S3V0), C (Q9S3U9), D (Q9S3U8), and E (Q7NSZ5), as well as target proteins using the RCSB protein database (PDB) (http://www.rcsb.org/, accessed on 20 January 2022). The docking-related parameters were set, and AutoDock 4.2 software (10550 North Torrey Pines Road, CA, USA) was used for molecular docking assessments. The signal molecule C10-Hls was used as a control to list the statistics of the binding energy of dietary phytochemicals and the CviR target protein for comparison and analysis.

### 4.8. Action of Active Compounds on vioABCDE Operon of C. violaceum

To further explore the possible mechanisms for inhibition of QS by active compounds from *Gnaphalium hypoleucum* DC. [6], qRT-PCR was performed to investigate the transcription level of the *vioABCDE* operon with and without active compounds, Control active compound concentrations of 31.25, 15.63, 7.81 µg/mL were determined, using rpoB (RNA polymerase subunit B) as the reference gene. Total RNA was extracted using a RNA prep Pure Cell/Bacteria Kit, according to the manufacturer’s guidelines. Then, the oligonucleotide primers were designed and synthesized by the Shanghai Shenggong company (Appendix A). Quantitative RT-qPCR was performed using the Real-Time PCR Master Mix (SYBR Green) and an ABI PRISM 7500 Real-time PCR system (Applied Biosystems) with two independent cultures. All experiments were performed in triplicate. The 2−ΔΔCt method was used to analyze the data of the quantitative real-time PCR.

### 4.9. Statistical Analysis

All experiments were performed at least in triplicate, and all data were analyzed using Graph Pad Prism 5. The data obtained from experiments were presented as mean values, and the difference between the control and tested groups were analyzed using Student’s *t*-test. The significant differences were *p* < 0.05.

## 5. Conclusions

In conclusion, flavonoid metabolites in *G. hypoleucum* DC. were predicted through metabolomics, and five flavonoid compounds were isolated from extracts and identified. Based on anti-QS activity and anti-biofilm tracking, apigenin and luteolin exhibited strong QS inhibitory effects on *C. violaceum* and could interfere with violacein biosynthesis by downregulating the *vioB*, *vioC*, and *vioD* genes. It was also found that violaceum cells will actively take up into the cells and combine with the target protein CviR to express the related *vioABCDE* gene, thereby exhibiting the characteristics of purple pigment production, cell membrane production, and migration. This is the first report to demonstrate that *G. hypoleucum* DC. has potential inhibitory activity against QS. Apigenin and luteolin could be used in the development of novel antimicrobial agents to treat pathogenic infections.

## Figures and Tables

**Figure 1 molecules-27-04881-f001:**
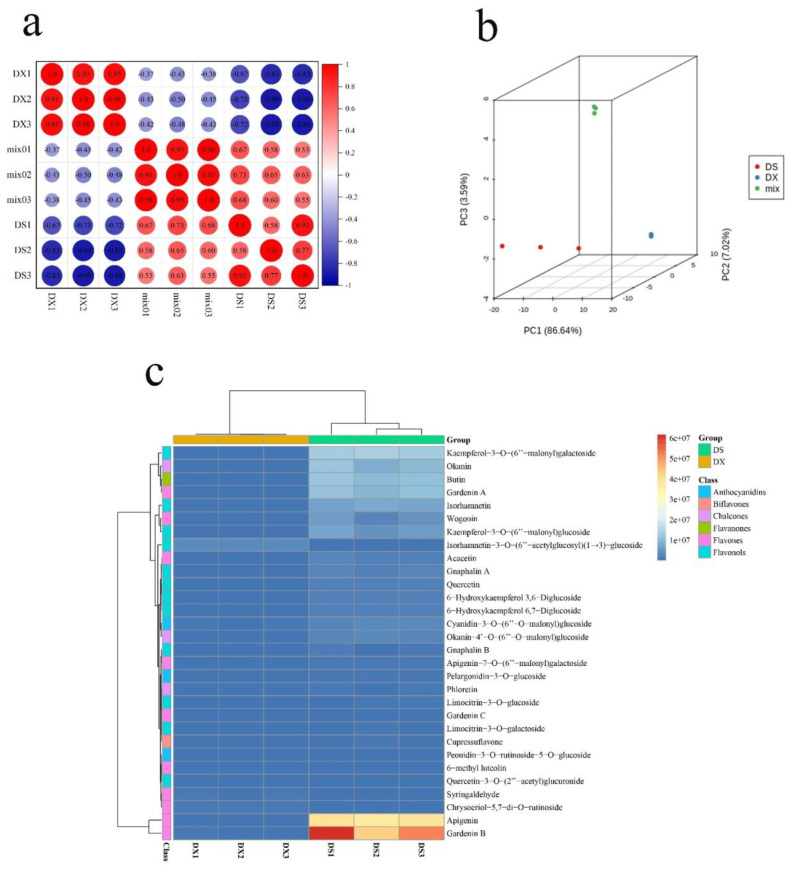
Heat map of main components in *Gnaphalium hypoleucum* DC. identified by metabolome (**a**) Correlation analysis of the *G*. *hypoleucum* DC. above-ground group (DS) and subterranean group (DX) samples, with the redder the color representing the greater correlation. (**b**) Principal component analysis of metabolomic data from above-ground (DS) and below-ground (DX) parts of *G*. *hypoleucum* DC. The “mix” refers to an equilibrium mixture of all samples (quality control) (**c**) A heat map of the identified compounds in *G*. *hypoleucum* DC. in the two groups—the colors indicates the relative content of different groups of compounds by the average peak response area by UPLC-MS, and a redder color indicates a higher relative content of the compound.

**Figure 2 molecules-27-04881-f002:**
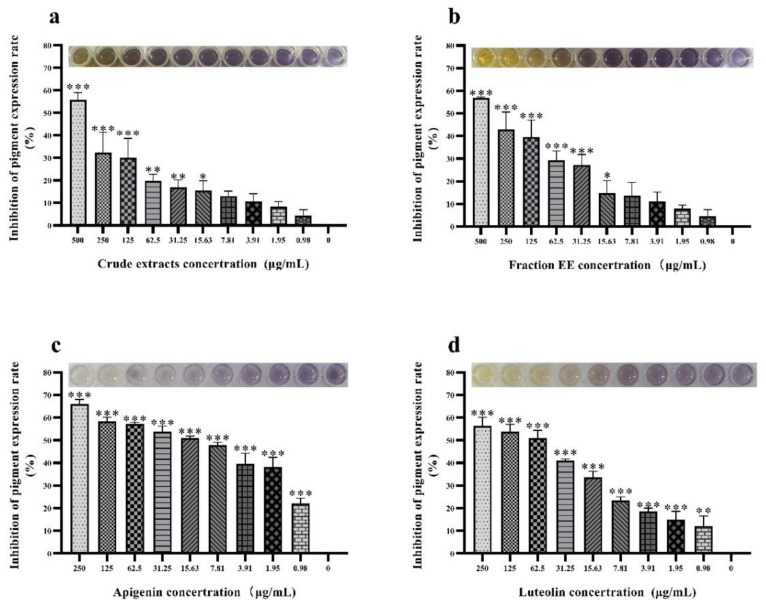
The *G. hypoleucum* DC. extract and two compounds isolated from it showed an inhibitory effect on violacein production by *C. violaceum*. In (**a**,**b**), violacein inhibition by *C. violaceum* was tested at OD_585_ in the presence of crude extracts and ethyl acetate partition (fraction EE) (mg/mL), respectively, at 30 °C after 24 h in 96-well plates. (**c**,**d**) Luteolin and apigenin separated from fraction EE significantly inhibited violacein production in a concentration-dependent manner, respectively. Inhibition of pigment expression rate was calculated using the following formula: Inhibition% = treated OD_585_/control OD_585_ × 100%. Columns represent means ± standard deviations, Statistically significant differences (determined by Student’s *t*-test) are indicated as *** *p* < 0.001, ** *p* < 0.01, and * *p* < 0.05 vs. the control group.

**Figure 3 molecules-27-04881-f003:**
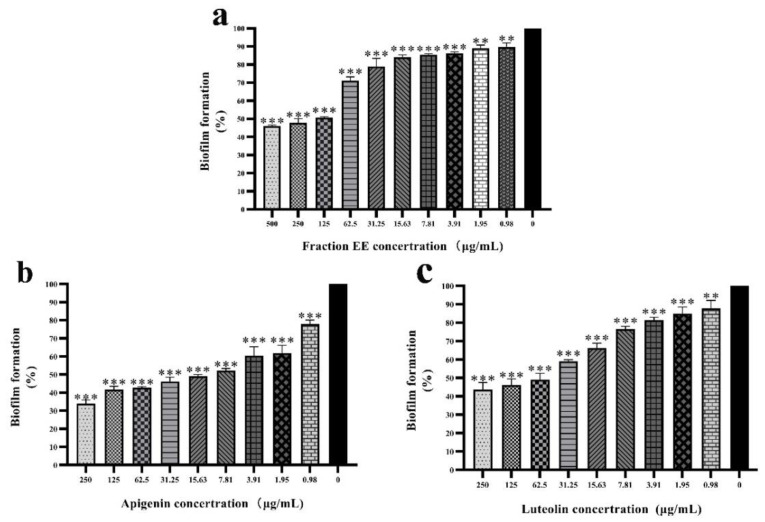
Inhibitory rate of apigenin, luteolin, and EE fraction on *C. violaceum* biofilm formation. (**a**) Biofilm formation of *C. violaceum* was quantified at OD_490_ in the presence of EE fraction at 0–500 µg/mL 30 °C after 24 h in 96-well plates. (**b**) Biofilm formation of *C. violaceum* was quantified at OD_490_ in presence of apigenin at 0~250 µg/mL 30 °C after 24 h in 96-well plates. (**c**) Biofilm formation of *C. violaceum* was quantified at OD_490_ in presence of luteolin at 0–250 µg/mL 30 °C after 24 h in 96-well plates. Columns represent means ± standard deviations. Statistically significant differences (determined by Student’s *t*-test) are indicated as *** *p* < 0.001, ** *p* < 0.01 vs. the control group.

**Figure 4 molecules-27-04881-f004:**
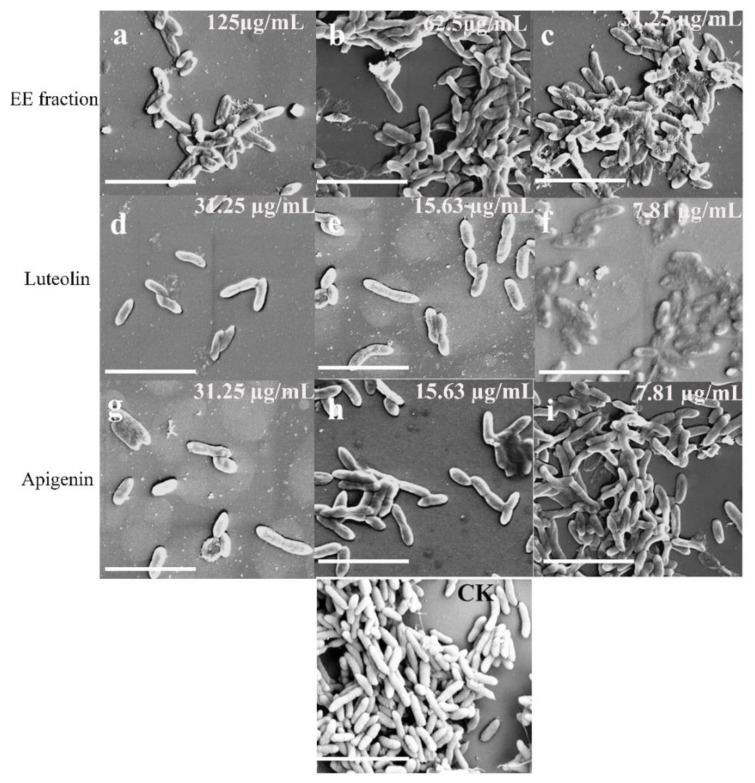
Inhibiting effect of apigenin, luteolin, and EE fraction on the biofilm formation of *C. violaceum* by SEM. (CK) Biofilm formation of *C. violaceum* was quantified at OD_490_ at mg/mL 30 °C after 24 h in 96-well plates. We aspirated the medium, used gradient ethanol and acetone to prepare for dehydration, and processed each gradient at 15 min intervals. The final zero boundary point drying occurred and the sample tank was sprayed gold, and finally SEM was used (variable vacuum ultra-high resolution field emission scanning electron microscope/energy spectrometer Thermo Fisher Waltham, MA, USA) to observe the sample photos (**a**–**c**) in the presence of EE fraction at 125, 62.5, and 31.25 µg/mL, (**d**–**f**) in the presence of luteolin at 31.25, 15.63, and 7.81 µg/mL, and (**g**–**i**) in the presence of apigenin at 31.25, 15.63, and 7.81 µg/mL. The bar is 5 μm.

**Figure 5 molecules-27-04881-f005:**
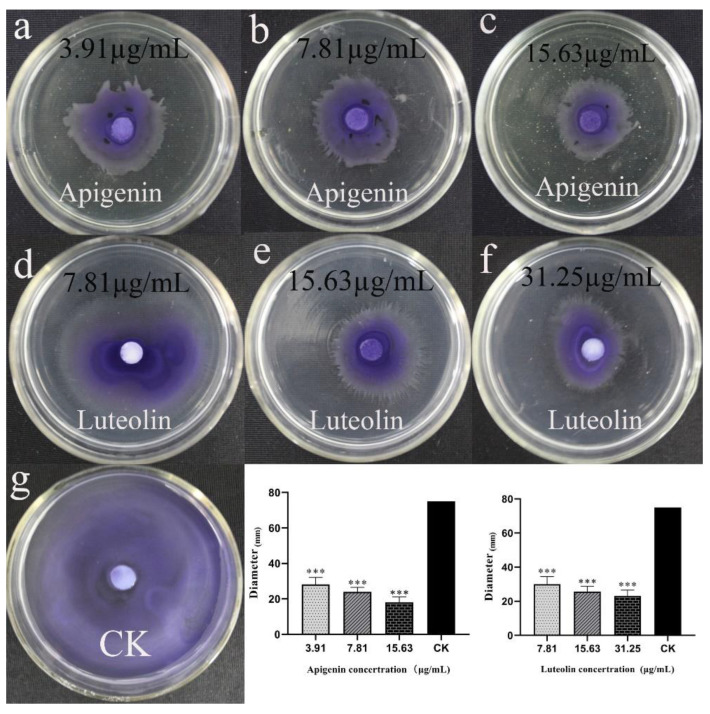
Effects of apigenin and quercetin on motility of *C. violaceum*. In the motility experiment, (**a**–**c**) are the apigenin treatment groups, the (**d**–**f**) groups are the luteolin treatment groups, and (**g**) is the control group. By comparing with the control group, as the drug concentration increases from left to right, we find that the apigenin group treatment inhibits swimming. The sexual effect changes more obviously, and the inhibitory ability of apigenin is stronger than of luteolin. Columns represent means ± standard deviation. Statistically significant differences (determined by Student’s *t*-test) are indicated as *** *p* < 0.001 vs. the control group. The diameter of the petri dish is 80 mm.

**Figure 6 molecules-27-04881-f006:**
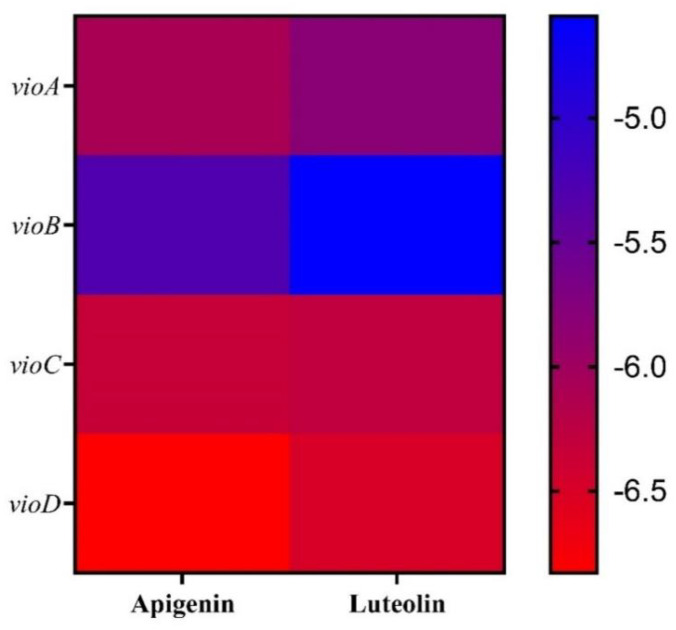
Molecular docking predicted energy heatmap. The figure uses different degrees of red to statistically display the binding energy of all docking prediction results. The redder the color, the lower the binding energy, and the higher the possibility of binding.

**Figure 7 molecules-27-04881-f007:**
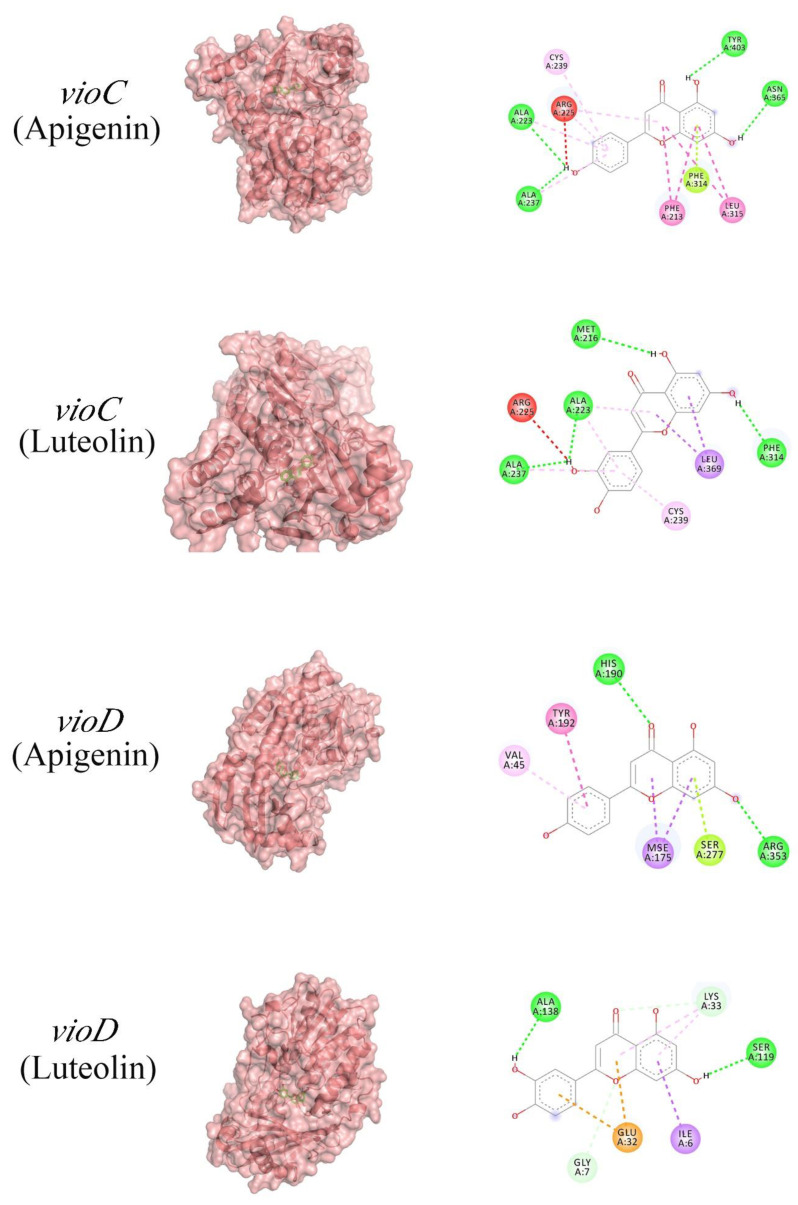
Molecular docking prediction map. The left side is the three-dimensional structure of the connection position, and the right side is the label of the specific binding site.

**Figure 8 molecules-27-04881-f008:**
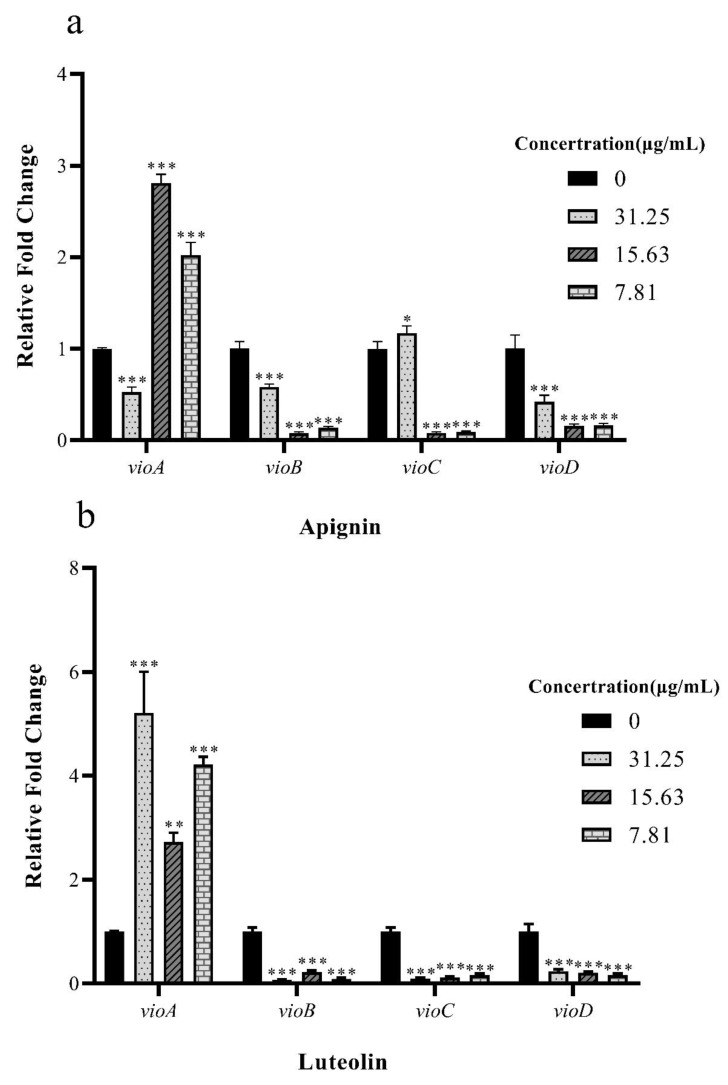
The ability of apigenin and luteolin to inhibit the expression of key genes of *C. violaceum* quorum sensing (**a**) is the apigenin group, and (**b**) is the luteolin group. We can find that both of them do not inhibit the expression of the *vioA* gene, and that luteolin has a relatively obvious effect on *vioB*, *vioC*, and *vioD* inhibition, while apigenin is not as good as luteolin in inhibiting the inhibition of the *vioC* gene. Columns represent means ± standard deviations. Statistically significant differences (determined by Student’s *t*-test) are indicated as *** *p* < 0.001, ** *p* < 0.01, and * *p* < 0.05 vs. the control group.

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
