# Peer review of "The Derived Components of *Gnaphalium hypoleucum* DC. Reduce Quorum Sensing of *Chromobacterium violaceum"

_molecules, 2022, doi:10.3390/molecules27154881_

Round 1

Reviewer 1 Report

The paper "The derived components of Gnaphalium hypoleucum DC. reduce the biofilm formation, motility and quorum sensing formation of C. violaceum ATCC 12472" by Yu-Long Li et all reports on the anti-bacterial activity of the plant Gnaphalium hypoleucum widely distributed in Yunnan province of China and used in the local medicine. It is an interesting paper that provides some valuable new insights into the research of this medicinal plant. Some critical comments must be first addressed before the paper could be considered for publication in Molecules.

Some typos and errors are met throughout the manuscript, e.g. in line 17 ("G, hypoleucum DC") or in line 95 ("C. violaceum 12472" should be italics).

The results section begins with "Metabolomics prediction results" which is not clear. What was predicted and how? What are the identified metabolites in the crude extract? What were the spectra and the related information of the identified compounds (their quality, quantity, etc.)? The data is missing in both the main text and supplementary. There is no information on how the samples were prepared and why, and therefore it is hard to follow what the aboveground (DS) and underground (DX) samples mean;

line 77. "samples with different treatments, and the stability within the group was good". 1) what do "different treatments" mean here? 2) what is considered by "good stability"?;

lines 72 and 82: some information would be more appropriate in the Materials, in particular, "Unsupervised PCA (principal component analysis) was performed by statistics function prcomp within R (www.r-project.org). The data was unit variance scaled before unsupervised PCA." and "Both HCA and PCC were carried out by R package pheat map." ;

line 84: The sentence "Metabolome prediction results indication that G. hypoleucum DC. including above ground (DS) and below ground" is hard to understand, please, check the sentence;

line 86. The full details on the identified compounds must be provided;

Figure 1. The labels in Figure 1 are hard to read, please resize the figure labels so that one can easily read them. The figure legend does not have the beginning;

line 94. MIC abbreviation should be given as full name at first mention;

line 197. how the biofilm inhibition was assessed in this part "In this study, the biofilm inhibition potential of EEfraction, apigenin and luteolin were evaluated at different concertration."?

line 283: gene names should be in italics.

Figure 5. Using two colors (e.g. yellow and blue) would make it easier to see the gradient between the minimal and maximal values.

Figure 3. The figure legend does not correspond to the figure itself. 

Figure 4. The figure legend does not have a beginning. the scale bars could be added to better characterize inhibition zones. Compound concentrations should be given either in a legend or in the figure (a-f).

English requires extensive editing to keep it to a standard while some sentences are hard to understand; e.g., the sentence in lines 589-590: "Evaluate the ability of drugs to suppress swarm-sensing effect by measuring the movement of the swarm by migration distance." or in line 617 "The oligonucleotide primers were designed and synthesis (Table S1)."

Literature and discussion. The introduction seems a bit short in reviewing the current state of biofilm research. You could discuss more research data on recent advances in various approaches to combat microbial biofilms, for instance using furanones, which are also known to interfere with quorum sensing and biofilm formation: https://pubmed.ncbi.nlm.nih.gov/30671584/ or https://pubmed.ncbi.nlm.nih.gov/32024254/

Some experimental details are missing in the Materials, for instance, the culture conditions of Gnaphalium hypoleucum DC. and C. violaceum ATCC 12472.

Lines 3, 22, and 39: full name should be provided at the first mention for "C. violaceum";

line 23 and 28: check if the "vio B, vio C, and vio D" gene names are correct;

line 37: "... and so on [1]" sounds too informal, consider rewriting the sentence; 

line 62: correct "metabonomics"

line 522: full name for the "UPLC-ESI-MS/MS" abbreviation is missing;

line 566: "Quantitative analysis of violacein production was conducted according to the methods reported in the literature [9]." Please provide a brief description of the analysis in the Materials.

lines 570-571: The following sentence is inappropriate in the Materials, better delete it or remove it elsewhere where appropriate: "Biofilm formation is one of vital form of multicellular behaviour about cell-to-cell communication and it has been shown to greatly contribute to bacterial pathogenesis."

line 580: what was the solvent for crystal violet to read the absorbance?

line 591: please, provide the details on how the swimming and swarming were analyzed.

Author Response

Responses to the reviewers

To Reviewer #1:

1.The paper "The derived components of Gnaphalium hypoleucum DC. reduce the biofilm formation, motility and quorum sensing formation of C. violaceum ATCC 12472" by Yu-Long Li et all reports on the anti-bacterial activity of the plant Gnaphalium hypoleucum widely distributed in Yunnan province of China and used in the local medicine. It is an interesting paper that provides some valuable new insights into the research of this medicinal plant. Some critical comments must be first addressed before the paper could be considered for publication in Molecules.

Some typos and errors are met throughout the manuscript, e.g. in line 17 ("G, hypoleucum DC") or in line 95 ("C. violaceum 12472" should be italics).

Responses: Thank you for your nice suggestion. We have checked and corrected all latin name as italics in the manuscript and marked them with red.

2.The results section begins with "Metabolomics prediction results" which is not clear. What was predicted and how? What are the identified metabolites in the crude extract? What were the spectra and the related information of the identified compounds (their quality, quantity, etc.)? The data is missing in both the main text and supplementary. There is no information on how the samples were prepared and why, and therefore it is hard to follow what the aboveground (DS) and underground (DX) samples mean;

Responses: Thanks to your suggestion, we have changed the results section with "Metabolomics prediction results" to Metabolomics prediction results of Gnaphalium hypoleucum DC. We will add spectrally relevant information for the extracted compounds in the supplementary. In the article, lines 309-303, Metabolomics analysis and active ingredient extraction of Gnaphalium hypoleucum DC. We explain why we should separate aboveground (DS) and underground (DX).

3.line 77. "samples with different treatments, and the stability within the group was good". 1) what do "different treatments" mean here? 2) what is considered by "good stability"?;

Responses: Thank you for your nice suggestion. We are not divided into different treatment groups here, but are grouped according to different parts of the plant, divided into aboveground (DS) and underground (DX). The stability of the data in Principal Component Analysis is actually to prove the grouping is the data reliable. We have deleted this sentence.

4.lines 72 and 82: some information would be more appropriate in the Materials, in particular, "Unsupervised PCA (principal component analysis) was performed by statistics function prcomp within R (www.r-project.org). The data was unit variance scaled before unsupervised PCA." and "Both HCA and PCC were carried out by R package pheat map." ;

Responses: Thanks for your suggestion, we have placed this information in the method and marked in red.

  1. line 84: The sentence "Metabolome prediction results indication that G. hypoleucum DC. including above ground (DS) and below ground" is hard to understand, please, check the sentence;

Responses: Thank you for your suggestion, we have revised the sentence in the manuscript as following Metabolome test results indication that G. hypoleucum DC. DSand DXhave apigenin, luteolin, 5,7-dihydroxy-3-methoxyflavone, aromadendrin-7-O-glucoside, peonidin-3-O-glucoside, kaempderide, okanin, quercetin and so on flavonoid components, It's worth noting that apigenin and luteolin content in DS section of G. hypoleucum DC. are higher compared with other flavonoids

  1. line 86. The full details on the identified compounds must be provided;

Responses: These compounds that we isolated from this plant were not new compounds and we offer the HPLC figure in supplementary.

7.Figure 1. The labels in Figure 1 are hard to read, please resize the figure labels so that one can easily read them. The figure legend does not have the beginning;

Responses: Thanks for your suggestion, we have revised the label of Figure 1

Figure 1 Heat map of main components and their contents in Gnaphalium hypoleucum DC.

Figure 2 G. hypoleucum DC. extract and two compounds isolated from it showed an inhibitory effect on violacein production by C. violaceum ATCC 12472.

Figure 3 Inhibiting effect of EEfraction, apigenin and luteolin on the biofilm formation of C. violaceum ATCC 12472.

Figure 4 Observed the effects of apigenin and quercetin on motility of C. violaceum ATCC12472 at concentrations that minimally inhibit pigment production and biofilm formation.

Figure 5 Molecular docking predicted energy heatmap.

Figure 6 Molecular docking prediction map.

Figure 7 The ability of apigenin and luteolin to inhibiting the expression of key genes of C. violaceum ATCC12472 quorum-Sensing.

8.line 94. MIC abbreviation should be given as full name at first mention;

Responses: Thank you for your nice suggestion, we have added MIC full name as Minimum inhibitory concentration at first mention.

9.line 197. how the biofilm inhibition was assessed in this part "In this study, the biofilm inhibition potential of EE fraction, apigenin and luteolin were evaluated at different concertration."?

Responses: Thanks, Bacterial biofilms formation beneficial to enhance it is tolerance to antibiotics, and environmental pressure. Besides, the formation of bacterial biofilms is regulated by the QS system. In this study, the biofilm inhibition potential of EE fraction, apigenin and luteolin were evaluated at different concentrations. The formation of biofilms increased with decreasing drug concentration in a classic dose-dependent response. At a concentration of 31.25 μg/mL, the biofilm inhibition rates were 46.15%, 59.05%, and 78.97% for apigenin, luteolin, and EE fraction, respectively.

10.line 283: gene names should be in italics.

Responses: I am sorry for my careless, We have checked and corrected all gene names as italics in the article and marked them with red.

11.Figure 5. Using two colors (e.g. yellow and blue) would make it easier to see the gradient between the minimal and maximal values.

Responses: Thanks for your suggestion, we have revised the color of Figure 5. Using red and green, we revised the sentence, marked in red.

12.Figure 3. The figure legend does not correspond to the figure itself. 

Responses: Thanks for your suggestion, we have modified Figure 3, and modified the figure legend, marked in red.

Figure 3. Inhibiting effect of apigenin and luteolin on the biofilm formation of C. violaceum ATCC 12472. (CK) Biofilm formation of C. violaceum12472 was quantified at OD490 at mg/mL 30 °C after 24 h in 96-well plates. Aspirate the medium, use gradient ethanol and acetone to prepare for dehydration, and process each gradient interval 15min, the final zero boundary point drying and the sample tank spray gold, and finally use SEM (Variable vacuum ultra-high resolution field emission scanning electron microscope/energy spectrometer Thermo Fisher USA) to observe the sample photos.a-cin presence of EE fraction at 31.25, 62.5, 125 μg/mL.(d-f)in presence of luteolin at 31.25, 15.63, 7.81 μg/mL.(g-i)in presence of apigenin at 31.25, 15.63, 7.81 μg/mL .The bar is 5μm. (j) Biofilm formation of C. violaceum12472 was quantified at OD490 in presence of EE fraction at 31.25, 62.5, 125 μg/mL 30 °C after 24 h in 96-well plates. (k)Biofilm formation of C. violaceum12472 was quantified at OD490 in presence of apigenin at 31.25, 15.63, 7.81 μg/mL 30 °C after 24 h in 96-well plates. (m) Biofilm formation of C. violaceum12472 was quantified at OD490 in presence of luteolin at 31.25, 15.63, 7.81 μg/mL 30 °C after 24 h in 96-well plates.

  1. Figure 4. The figure legend does not have a beginning. the scale bars could be added to better characterize inhibition zones. Compound concentrations should be given either in a legend or in the figure (a-f).

Responses: Thanks to your suggestion, we have changed Figure 4 and added a description of the legend, marked in red.

Figure 4. Effects of apigenin and quercetin on motility of C. violaceum ATCC12472.

In the motility experiment, a, b, and c are the apigenin treatment groups, the d, e, and f groups are the luteolin treatment group, and g is the control group. By comparing with the control group, As the drug concentration increases from left to right, we find that the apigenin group treatment inhibits swimming. The sexual effect changes more obviously, and the inhibitory ability of apigenin is stronger than of luteolin. The bar is 2cm.

  1. English requires extensive editing to keep it to a standard while some sentences are hard to understand; e.g., the sentence in lines 589-590: "Evaluate the ability of drugs to suppress swarm-sensing effect by measuring the movement of the swarm by migration distance." or in line 617 "The oligonucleotide primers were designed and synthesis (Table S1)."

Responses: Thanks for your suggestion, we have revised the sentence and marked it in red.

The line 617 "The oligonucleotide primers were designed and synthesis” have revised as following: Then the oligonucleotide primers were designed and synthesized by Shanghai shenggong company (Table S1).

  1. Literature and discussion. The introduction seems a bit short in reviewing the current state of biofilm research. You could discuss more research data on recent advances in various approaches to combat microbial biofilms, for instance using furanones, which are also known to interfere with quorum sensing and biofilm formation: https://pubmed.ncbi.nlm.nih.gov/30671584/ or https://pubmed.ncbi.nlm.nih.gov/32024254/

Responses: Thanks for your nice suggestion, This literature has been cited in the discussion section as following: Furanone and furanone derivative have a good inhibition effect of biofilm formation, such as food-poisoning bacterium Bacillus cereus, Staphylococcus aureus, Staphylococcus epidermidis, Micrococcus luteus.

  1. Some experimental details are missing in the Materials, for instance, the culture conditions of Gnaphalium hypoleucum DC. and C. violaceum ATCC 12472.

Responses: Thanks for your suggestion, Gnaphalium hypoleucum DC. is the plant material we collected directly in the natural environment, so there is no relevant culture data. The culture data of C. violaceum ATCC 12472. We presented in the article 338-359, cultured at 30°C for 16h, the relevant paragraphs are marked in red.

17.Lines 3, 22, and 39: full name should be provided at the first mention for "C. violaceum";

Responses: Thanks for your suggestion, we have revised the phrase and it is marked in red.

18.line 23 and 28: check if the "vio B, vio C, and vio D" gene names are correct;

Responses: Thanks for the suggestion, we have checked the gene name to confirm it is correct.

19.line 37: "... and so on [1]" sounds too informal, consider rewriting the sentence; 

Responses: Thanks for your suggestion, we have revised the sentence and marked it in red

20.line 62: correct "metabonomics"

Responses: Thanks for your suggestion, we have revised the word and it is marked in red

21.line 522: full name for the "UPLC-ESI-MS/MS" abbreviation is missing;

Responses: Thanks, the full name of "UPLC-ESI-MS/MS" is “ultra-performance liquid chromatography electrospray ionization mass spectrometry”.

  1. line 566: "Quantitative analysis of violacein production was conducted according to the methods reported in the literature [9]." Please provide a brief description of the analysis in the Materials.

Responses: Thanks for your suggestion, we have provide a brief description of the analysis in the Materials, as following: Aspirate 200 µL of the different treatment groups's culture in a 96-well plate from each culture and adding the same amount of ethyl acetate. The mixture was then shaken at maximum speed by a vortex apparatus for 2 min and centrifuged at 6000 g for 1 min. The organic phase was recovered and absorbance was read with a spectrophotometer at a wavelength of 575 nm. To calculate the percentage of inhibition, absorbance of the controls was considered to be 100% production of violacein. Each experiment was performed with three independent cultures.

23.lines 570-571: The following sentence is inappropriate in the Materials, better delete it or remove it elsewhere where appropriate: "Biofilm formation is one of vital form of multicellular behaviour about cell-to-cell communication and it has been shown to greatly contribute to bacterial pathogenesis."

Responses: Thanks for your nice suggestion, we have deleted this sentence.

  1. line 580: what was the solvent for crystal violet to read the absorbance?

Responses: Thank you for your suggestion. In the crystal violet staining biofilm formation experiment, we did not use other solvents for reading, but directly read the washed well plate.

25.line 591: please, provide the details on how the swimming and swarming were analyzed.

Responses: Thanks for your suggestion. In our article 2.5 we explained how to evaluate swimming, we quantitatively test the swimming ability of C. violaceum ATCC 12472 by measuring the swimming distance between the experimental group and the control group.

Reviewer 2 Report

The derived components of Gnaphalium hypoleucum DC. Reduce the biofilm formation, motility and quorum sensing formation of C. violaceum ATCC 12472

In this work, crude extracts of G. hypoleucum DC. contained significant amounts of two compounds shown to inhibit bacterial QS activity: apigenin and luteolin. Apigenin and luteolin in crude extracts of G. hypoleucum DC. showed substantial inhibition of pigment formation, biofilm production, and motility in C. violaceum ATCC 1247 compared to the effects of other phytochemicals from G. hypoleucum DC.

Introduction

In the introduction it is necessary to mention the chemical background of the plant.

Results

Figure 1 has no caption. PCA for 2 components.

All the figures are misplaced, please check that they do not delete text, that they have their figure caption and the resolution.

In Molecular docking prediction It is necessary to describe the interactions by hydrogen bridge and the hydrophobic ones, in the supplementary material, put it in tables and in the results highlight the most important interactions. Calculate the equilibrium constant Ki.

4.6 Molecular docking prediction

It is necessary to mention the size of the box and the center of the coordinates used for the calculations, in addition they do not mention anything about the validation of the proteins. Validation is required.

In the conclusion it does not mention the coupling, the molecular biology... this is weak, strengthen it with found experimental data.

Author Response

Reviewer2:

The derived components of Gnaphalium hypoleucum DC. Reduce the biofilm formation, motility and quorum sensing formation of C. violaceum ATCC 12472

In this work, crude extracts of G. hypoleucum DC. contained significant amounts of two compounds shown to inhibit bacterial QS activity: apigenin and luteolin. Apigenin and luteolin in crude extracts of G. hypoleucum DC. showed substantial inhibition of pigment formation, biofilm production, and motility in C. violaceum ATCC 1247 compared to the effects of other phytochemicals from G. hypoleucum DC.

Introduction

In the introduction it is necessary to mention the chemical background of the plant.

 Responses: Thanks for your nice suggestion. We have added chemical background of the plant in the introduction. As following: Phytochemical compounds in the plant are flavonoids as major constituents and that most of them are 5-OH flavanones, aurantiamide acetate, 5-hydroxy-3,6,7,8-tetramethoxyflavone, 5-hydroxy-3,6,7,8,4’-pentamethoxy flavones, tetracosanoic acid, β-sitosterol,5-hydroxy-3,6,7,8,3’,4’-hexa methoxyflavone, 5,7-dihydroxy- 3,8,4’-trimethoxyflavone, 5,8-Dihydroxy-3,6,7-trimethoxyflavone

Results

Figure 1 has no caption. PCA for 2 components.

All the figures are misplaced, please check that they do not delete text, that they have their figure caption and the resolution.

Responses: Thanks for your suggestion, we have revised the label of Figure 1, Besides, other Figure caption have also added in revised manuscript.

Figure 1 Heat map of main components and their contents in Gnaphalium hypoleucum DC.

Figure 2 G. hypoleucum DC. extract and two compounds isolated from it showed an inhibitory effect on violacein production by C. violaceum ATCC 12472.

Figure 3 Inhibiting effect of EEfraction, apigenin and luteolin on the biofilm formation of C. violaceum ATCC 12472.

Figure 4 Observed the effects of apigenin and quercetin on motility of C. violaceum ATCC12472 at concentrations that minimally inhibit pigment production and biofilm formation.

Figure 5 Molecular docking predicted energy heatmap.

Figure 6 Molecular docking prediction map.

Figure 7 The ability of apigenin and luteolin to inhibiting the expression of key genes of C. violaceum ATCC12472 quorum-Sensing.

In Molecular docking prediction It is necessary to describe the interactions by hydrogen bridge and the hydrophobic ones, in the supplementary material, put it in tables and in the results highlight the most important interactions. Calculate the equilibrium constant Ki.

Responses: The linking group of the hydrogen bridge, we have mentioned in the article 2.6, after we predicted the binding ability of the compound to the gene by molecular docking, we also verified it by qRT-PCR, so we think it is not necessary to calculate the KI to help prove the molecule butt

 4.6 Molecular docking prediction

It is necessary to mention the size of the box and the center of the coordinates used for the calculations, in addition they do not mention anything about the validation of the proteins. Validation is required.

 Responses: Thanks for your suggestion, we have added Table S4 Detailed data for calculations in molecular docking in the supplementary material.

 In the conclusion it does not mention the coupling, the molecular biology... this is weak, strengthen it with found experimental data.

Responses: Thanks, We have added as much knowledge as possible according to the experimental results. As following:

In conclusion, flavonoid metabolites in G. hypoleucum DC. were predicted through metabolomics, and five flavonoid compounds were isolated from extracts and identified. Based on anti-QS activity and anti-biofilm tracking, apigenin and luteolin exhibited strong QS inhibitory effects on C. violaceum ATCC12472 and could interfere with violacein biosynthesis by downregulating the vio B, vio C, and vio D genes and violaceum cells will actively take up into the cells and combine with the target protein CviR to express the related vio A, B, C, D, E gene, thereby exhibiting the characteristics of purple pigment production, cell membrane production and migration. This is the first report to demonstrate that G. hypoleucum DC. has potential inhibitory activity against QS. Apigenin and luteolin could be used in the development of novel antimicrobial agents to treat pathogenic infections.

Round 2

Reviewer 1 Report

The revised manuscript has some improvements , however is still far from recommending it to publish. Not all the comments were addressed. The written English is far from the standards with a plenty of grammatical errors and typos. The detailed responses are in a point-by-point manner:

1) There are still some incorrectly written species names (not in italics)

2) I did not find an explanation for what DS and DX are. Authors responded that this information can be found in lines 309-303, however it is not the case. Line 74: An introductory sentence is needed to understand what are "aboveground (DS) and underground (DX) samples" and why the samples were analyzed separately.

Furthermore, "Metabolomics prediction results" seem incorrect to me. It is rather "Identification" of metabolites and not the prediction.

3) If you used different parts, then you should write "parts" and not the "treatments".

5) The revised sentence is still not readable: Metabolome test results indication that G. hypoleucum DC. DSand DXhave apigenin, luteolin, 5,7-dihydroxy-3-methoxyflavone, aromadendrin-7-O-glucoside, peonidin-3-O-glucoside, kaempderide, okanin, quercetin and so on flavonoid components, It's worth noting that apigenin and luteolin content in DS section of G. hypoleucum DC. are higher compared with other flavonoids”. English editing is required.

9) The question for line 197 was: "how the biofilm inhibition was assessed?". Please add the information on what the method was used for assessing anti-biofilm activity.

12) Figure 3 is hard to read. I suggest to make two separate figures, one with SEM images (a-i) and the other with anti-biofilm activity graphs (j,k,m). It would be also better to write the compounds and concentrations in the figure itself next to the microscopy images.

  1. Please, provide the details how the plant was collected in the Materials, give as much details as possible. Maybe here you could write why you analyzed different plant parts.

18) I believe that the "vio B, vio C, and vio D" gene names should be written as  "vioB, vioC, and vioD". 

  1. line 580: The authors write that "In the crystal violet staining biofilm formation experiment, we did not use other solvents for reading, but directly read the washed well plate." In order to read the absorbance, the crystal violet must be solubilized in ethanol or equivalent. It is written in the referenced work or others:

    https://www.frontiersin.org/articles/10.3389/fmicb.2017.01675/full

    https://www.frontiersin.org/articles/10.3389/fmicb.2017.02246/full

Other comments:

Lines 2-3: "biofilm formation" in the title is okay, but "quorum sensing formation" is not. Please, re-write it or simply leave "quorum sensing". The strain name "Chromobacterium violaceum ATCC 12472" is not very important in the title. I suggest making the following title (or similar): "The derived components of Gnaphalium hypoleucum DC. reduce the biofilm formation, motility and quorum sensing of Chromobacterium violaceum"

Line 22: It is enough to indicate the strain name at first mention (line 21) "Chromobacterium violaceum ATCC 12472". Delete "ATCC 12472" on lines 22, 27, 31, 41 and further where appropriate.

Lines 108 and 256: Provide the full name for "QSIs" abbreviation.

Figures 2 and 3 legends are separated from the corresponding figures by a text. Could you format that?

Some data discussed in lines 310-313 is not present in the results, in particular, "molecular docking results indication that main component of apigenin and luteolin had a docking score of -5.8 kcal/mol and -5.46 kcal/mol with CviR protein, respectively (Table S3). We specifically found that apigenin and luteolin were well placed in the 312 AHL binding pockets of the CviR protein of C. violaceum ATCC 12472." I haven't seen these results at all. This should be either deleted (also from the conclusions), or supported by the data in Results.

Author Response

The revised manuscript has some improvements , however is still far from recommending it to publish. Not all the comments were addressed. The written English is far from the standards with a plenty of grammatical errors and typos. The detailed responses are in a point-by-point manner: 1) There are still some incorrectly written species names (not in italics) Responses: I am sorry for my carelessness. We rechecked and changed all species names and marked them in blue. 2) I did not find an explanation for what DS and DX are. Authors responded that this information can be found in lines 309-303, however it is not the case. Line 74: An introductory sentence is needed to understand what are "aboveground (DS) and underground (DX) samples" and why the samples were analyzed separately. Responses: Thanks, We use metabolomics to identify precisely the chemical composition in G. hypoleucum DC. and compare the chemical composition of the aboveground (DS) and underground (DX) parts of plants. 3) Furthermore, "Metabolomics prediction results" seem incorrect to me. It is rather "Identification" of metabolites and not the prediction. Responses: Thank you for your suggestion. "Metabolomics prediction results" have revised as “Metabolome identification results of Gnaphalium hypoleucum DC.” 4) If you used different parts, then you should write "parts" and not the "treatments". Responses: Thank you for your suggestion. We have revised “treatments” as “parts” in revised manuscript. 5) The revised sentence is still not readable: “Metabolome test results indication that G. hypoleucum DC. DSand DXhave apigenin, luteolin, 5,7-dihydroxy-3-methoxyflavone, aromadendrin-7-O-glucoside, peonidin-3-O-glucoside, kaempderide, okanin, quercetin and so on flavonoid components, It's worth noting that apigenin and luteolin content in DS section of G. hypoleucum DC. are higher compared with other flavonoids”. English editing is required. Responses: Thanks, This sentence have revised as following:“Metabolome results of indication that apigenin, luteolin, 5,7-dihydroxy-3-methoxyflavone, aromadendrin-7-O-glucoside, peonidin-3-O-glucoside, kaempderide, okanin, quercetin and flavonoid components are identified in DS and DX parts of G. hypoleucum DC. It's worth noting that apigenin and luteolin’s content are higher in the DS part of G. hypoleucum DC. ” 6) The question for line 197 was: "how the biofilm inhibition was assessed?". Please add the information on what the method was used for assessing anti-biofilm activity. Responses: Thank you for your suggestion, we present what the method was used for assessing anti-biofilm activity in article 4.3. “Then the absorbance was determined at 595 nm [4]. Inhibitor-mediated reduction of biofilm formation was assessed by comparing it to the positive control without phytochemicals. The biofilm assay was performed at least three times.” 7) Figure 3 is hard to read. I suggest to make two separate figures, one with SEM images (a-i) and the other with anti-biofilm activity graphs (j,k,m). It would be also better to write the compounds and concentrations in the figure itself next to the microscopy images. Responses: Thank you for your suggestion, we have revised Figure 3. 8) Please, provide the details how the plant was collected in the Materials, give as much details as possible. Maybe here you could write why you analyzed different plant parts. Responses: Thanks for your suggestion, we have added sentences at 4.1 to explaining how the plant was collected and why we analyzed different plant parts. And we marked it in blue. “The whole plant of G. hypoleucum DC. were purchased from market in Yunnan province and were identified by Pro. Yanping Liu (NO. YNN20190008), and then, the specimens have been deposited in school of Basic Medicine, You jiang Medical University for Nationalities. In order to make the statistical results more accurate. We randomly sampled G. hypoleucum DC. 1 kg with similar phenotypic trait vigor and height. They were divided into 2 groups of above-ground and underground parts, and the stems, leaves, and flowers were selected as the above-ground parts (DS), and the roots were selected as the underground parts (DX). Three biological replicates were set in each group.” 9) I believe that the "vio B, vio C, and vio D" gene names should be written as "vioB, vioC, and vioD". Responses: Thank you for your suggestion, we have changed the writing format of the above nouns and marked them in blue. 10) line 580: The authors write that "In the crystal violet staining biofilm formation experiment, we did not use other solvents for reading, but directly read the washed well plate." In order to read the absorbance, the crystal violet must be solubilized in ethanol or equivalent. It is written in the referenced work or others: https://www.frontiersin.org/articles/10.3389/fmicb.2017.01675/full https://www.frontiersin.org/articles/10.3389/fmicb.2017.02246/full Responses: Thank you for your suggestion, we checked the method, added the sentence and marked it in blue “Then add 95% ethanol solution to the well, dissolve CV at 37°C for 30 minutes, determined it at 470 nm” Other comments: 11) Lines 2-3: "biofilm formation" in the title is okay, but "quorum sensing formation" is not. Please, re-write it or simply leave "quorum sensing". The strain name "Chromobacterium violaceum ATCC 12472" is not very important in the title. I suggest making the following title (or similar): "The derived components of Gnaphalium hypoleucum DC. reduce the biofilm formation, motility and quorum sensing of Chromobacterium violaceum". Responses: Thanks, the title have revised as “The derived components of Gnaphalium hypoleucum DC. reduce quorum sensing of Chromobacterium violaceum ”. 12)Line 22: It is enough to indicate the strain name at first mention (line 21) "Chromobacterium violaceum ATCC 12472". Delete "ATCC 12472" on lines 22, 27, 31, 41 and further where appropriate. Responses: Thank you for your suggestion, we have deleted "ATCC 12472" on lines 22, 27, 31, 41Lines 13) 108 and 256: Provide the full name for "QSIs" abbreviation. Responses: Thank you for your suggestion, we provide the full name for "QSIs" abbreviation on line 43 of the article, after that we use the abbreviation QSIs, we marked it in blue. Figures 2 and 3 legends are separated from the corresponding figures by a text. Could you format that? Responses: Thank you for your suggestion, we checked all the legends in the article and formatted them. Some data discussed in lines 310-313 is not present in the results, in particular, "molecular docking results indication that main component of apigenin and luteolin had a docking score of -5.8 kcal/mol and -5.46 kcal/mol with CviR protein, respectively (Table S3). We specifically found that apigenin and luteolin were well placed in the 312 AHL binding pockets of the CviR protein of C. violaceum ATCC 12472." I haven't seen these results at all. This should be either deleted (also from the conclusions), or supported by the data in Results. Responses: Thanks, We have added FigureS3 in supplementary.

Reviewer 2 Report

The work improved a lot with the modifications made, the graphs, figures and the supplementary information are suitable for publication. Thanks for making the changes.  

Author Response

The work improved a lot with the modifications made, the graphs, figures and the supplementary information are suitable for publication. Thanks for making the changes.  

Responses: Thanks.
